# A Multi-Objectives Genetic Algorithm Based Predictive Model and Strategy Optimization during SLM Process

**DOI:** 10.3390/ma15134607

**Published:** 2022-06-30

**Authors:** Qingfeng Xia, Jitai Han

**Affiliations:** 1School of Automation, Wuxi University, Wuxi 214105, China; xqf@cwxu.edu.cn; 2School of Management and Engineering, Nanjing University, Nanjing 210093, China; 3State Key Laboratory of Robotics, Shenyang Institute of Automation, Chinese Academy of Sciences, Shenyang 110016, China; 4School of Mechanical Engineering, Jiangnan University, Wuxi 214122, China

**Keywords:** multi-objectives genetic algorithm, selective laser melting, overhanging surface quality

## Abstract

Selective laser melting (SLM) process was optimized in this work using multi-objectives genetic algorithm. Process parameters involved in the printing process have an obvious impact on the quality of the printed parts. As the relationship between process parameters and the quality of different parts are complex, it is quite essential to study the effect of process parameter combination. In this work, the impact of four main process parameters, including defocusing amount, laser power, scan speed and layer thickness, were studied on overhanging surface quality of the parts with different inner structures. A multiple-factor and multiple-level experiment was conducted to establish a prediction model using regression analysis while multi-objective genetic algorithm was also employed here to improve the overhanging surface quality of parts with different inner shapes accordingly. The optimized process parameter combination was also used to print inner structure parts and compared with the prediction results to verify the model we have obtained before. The prediction results revealed that sinking distance and roughness value of the overhanging surface on a square-shape inner structure can reduce to 0.017 mm and 9.0 μm under the optimal process parameters combination, while the sinking distance and roughness value of the overhanging surface on a circle-shape inner structure can decrease to 0.014 mm and 10.7 μm under the optimal process parameters combination respectively. The testing results showed that the error rates of the prediction results were all within 10% in spite of random powder bonding in the printing process, which further proved the reliability of the previous results.

## 1. Introduction

Inner structure part has attracted increasing attention due to the fact that it can both decrease energy output ratio and increase the property of the printed part. However, its development has been seriously limited due to its difficulty in machining using a traditional manufacturing method.

Now, many researchers have refocused on inner structure part due to the common use of selective laser melting (SLM), which is considered one of the most promising additive manufacturing technologies. Compared to traditional manufacturing methods, SLM is different from typical subtractive manufacturing processes, and in theory can print parts with any complex structures. Therefore, this technology is popular with various applications such as biomedical and aerospace [1,2,3].

SLM-ed parts were printed under inert gas condition to prevent metal oxidation in the forming process. The powder was provided from a powder supplier with a roller and pushed towards the substrate for melting. Then a laser system scanned the powder on the substrate according to the 3D model and parameters set before. After that, the lifting plate under the substrate went down one-layer height while the lifting plate in supplier tank went up one-layer height. The above steps repeated until the model was printed [4,5,6].

However, the quality of the inner structure is hard to guarantee due to the lack of support in the printing process, which led to two main methods to improve the inner structure quality in the printing process: non-support printing and with-support printing. Researchers who considered non-support printing as a better choice tried to adjust a more suitable process parameter combination to increase the inner structure quality. Joel de Jesus et al. [7] determined that fatigue behavior is strongly affected by internal surface roughness, mainly in components manufactured by SLM. As compared with solid specimens, the surface roughness is the main cause of this fatigue strength reduction. Eren Pehlivan et al. [8] compared two post-processing method to improve the quality of the porous structure of the parts printed using SLM. He found that surface etching was a more effective way to increase the porous quality compared to hot isostatic pressing. Wang Di et al. [9] found that overhanging surface quality showed the most significant impact on the quality of the printed parts and optimized the overhanging surface quality by changing the inclination angle. Hongyu Chen et al. [10] tried to optimize the overhanging surface quality by adjusting relative process parameters and found that as an optimal processing parameter (60~80 J/mm^3^) was settled, the overhanging structure obtained a relatively smooth downward-facing surface due to the sound melt pool dimension and steady melt flow behavior. Jason C. Fox et al. [11] found that beam power, beam velocity, and overhanging angle all affected the overhanging surface quality. Jianbin Lu et al. [12] showed that at a smaller inclined angle and lower scan speed more serious warpage would happen, and the theoretical minimum building angle and reliable building angle fit with the experimental results at high and low scanning speed. AE Patterson et al. [13] used finite element analysis to develop the change of the overhanging features in SLM process caused by different parameters instead of practice-based setting and experiments were conducted to cerify the results gained in this work and fully demonstrate the reliability of the previous results. Jiang et al. [14] found that the laser surface energy density had a significant impact on the lower overhanging surface quality. They demonstrated that excessive energy density led to obvious sinking of the molten pool and a serious slag hanging phenomenon while too low energy density easily contributed to the insufficient powder fusion in the lower surface area, which led to the agglomeration of a molten pool during core processing, resulting in slag hanging, pores and powder spalling that reduced the quality of the lower surface.

Some researchers tried to print inner structure with support structure. Kajima et al. [15] studied the effect of adding support structure to fabricate overhanging surface. Results revealed that fatigue strength of overhanging surface printed with support structures was much better compared to that printed without support. It was mainly caused by the distortion reduction and increasing cooling rate of the overhanging layer printed with support. Zhang et al. [16] added cuboids into the conventional block type support structure and the Taguchi method was also applied to optimize support structure. Testing results revealed that the distortion of the sample was well controlled with this new support structure. Leary et al. [17] used voxel-based cellular automata method as fundamental to generate support structures in the printing process. He found that with these CA, it was possible to apply topology optimization geometries in the AM process. Zhang et al. [18] used branch-type support structure to replace traditional lattice-type support structure. He found that this new structure can both achieve cost-saving and strength-increasing in the printing process. Song et al. [19] used finite element analysis to minimize the residual stress and distormation of the overhanging structures with different support thickness. The testing results showed a good accordance with the prediction results which further proved the reliability of this work. Bartsch et al. [20] optimized the topology of the support structure using a combining process simulation which reduced the manufacturing and finishing efforts in the printing process using this method.

As the support structure was hard to remove in small inner holes, non-support printing process was employed in this work to print circle-shape and square-shape inner structures. Four main process parameters, including defocusing amount, laser power, scan speed and layer thickness were studied in this work. One thing should be noted is that these parameters were chosen on the basis of our previous study which had proved to have a significant impact on inner structure quality. An optimal process parameter combination was obtained to improve circle-shape and square-shape overhanging surface quality respectively using multi-objective genetic algorithm [21,22,23]. The different formation mechanism of square-shape and circle-shape inner structure was firstly discussed to explain the phenomenon we have gained from the experiment section as far as we know. One thing that should be noted is that as different SLM machine had different properties, only the equation and relative optimal process parameter combination can be used in this work, while the formation mechanism of differently shaped inner structure had versatility. Experiments were also conducted to verify the prediction model and previous results.

## 2. Experiment

### 2.1. Material

TC4 powder used in this work was provided by Shenzhen Minatech Co., Ltd., Shenzhen, China. The powder was first processed using ball milling machine supplied by PQ-N04, Across International CO., Ltd., Livingston, NJ, USA under 800 rpm rotating speed in both a clockwise and anticlockwise direction for 30 min, respectively. Then the powder was put into a tube furnace to dry for 50 min at 105 °C. The relative information about the after-processing powder is shown in Table 1 and Figure 1.

### 2.2. Instrument and Experiment

The SLM machine used in this work was provided by NUAA. This machine was designed and fabricated by using FS271M (Farson, Changsha, China). The schematic diagram of SLM can be seen in Figure 2.

The scan strategy used in this work was a Z-shape scan strategy as shown in Figure 3. Defocusing amount, laser power, scan speed and layer thickness were the four main parameters changed within the range we gained in our previous study, and other parameters were all kept the same in this work as shown in Table 2. To prevent random bonding of the unmelted powder in the inner structure to affect the reliability of the measured data, each sample had three feature structures and the designed model can be seen in Figure 4.

After printing, the samples were cut from the center to expose overhanging surfaces using Low-speed Wire Cutting machine provided by Suzhou BMG Precision Machinery Co., Ltd., Suzhou, China with the help of Wuxi Institute of Technology (Wuxi, China). To measure the sinking distance of the overhanging surfaces, Trilinear Coordinates Measuring instrument, Hexagon Metrology, Eskilstuna, Sweden, was also used in this work. Triangular laser measuring technique was employed by this instrument and the scanner head used here was HP-L-20.8 which had a working distance of 180 ± 40 mm. The scanning frequency was 100 Hz while the shape error was within 9 μm. Overhanging surface roughness was measured by Roughometer, Mitutoyo, Japan. The sampling length was taken as 2.5 mm in this work while the accelerating and decelerating length was 1.25 mm, respectively. The interval number was 5 and the length between each interval was 1 μm. To verify the measured data and have a better understanding on the formation of overhanging surfaces, a Scanning Electron Microscope (SEM) provided by Carl Zeiss, Sigma 300, Jena, Germany, was also used in this work to give an explanation. The acceleration voltage was 20 KV and its working distance was 8.7 mm. The magnification used in this work was 42× while the detector of this instrument was SE2.

## 3. Results and Discussion

As the single-factor experiment cannot describe the relationship between different process parameters, surface-response method was employed in this work and the mathematical model was established accordingly to analyze the overall influence caused by parameter combination. The relative expression used in this work was as follows:(1)$$y=f(x_{1}+x_{2}+\cdots +x_{p})+\omega$$

In this equation, x represents influence factor, y represents the response caused by these factors and ω represents the error term. Taking calculation speed and precision into account, a quadratic response surface regression model was used in this work which can be expressed as follows:(2)$$Y=\alpha_{0}+\sum_{i=1}^{m}\alpha_{i}x_{i}+\sum_{i=1}^{m}\alpha_{ii}x_{i}{}^{2}+\sum \sum_{i<j}\alpha_{ij}x_{i}x_{j}+\omega_{i}$$

In this equation, Y represents objective function, $\alpha_{0}$ represents constant term, $\alpha_{i}$ represents linear regression coefficient, $x_{i}$ and $x_{j}$ represent function argument, $\alpha_{ii}$ represents quadratic regression coefficient, $\alpha_{ij}$ represents interaction term regression coefficient, and $\omega_{i}$ represents the error term.

Based on our previous work, multi-factor and multi-level experiment was conducted within the range of process parameters listed in Table 3.

According to the factor-level set in Table 3, process parameter combinations were generated using optimal design in Response Surface section by Design-Expert software. The numeric factors used in this work was four while the categoric factors was zero. The type of each section was discrete while the levels used in this work was three. The model points in Runs section were 19 and the estimated lack of fit was 5. The total runs in this work was 29. The measured data for circle-shape and square-shape overhanging surface quality are listed in Table 4 and Table 5, respectively.

The mathematical prediction model on sinking distance and overhanging surface roughness of square-shape and circle-shape inner structure were calculated accordingly as shown in Equations (3)–(6). The analysis of variance in regression results of the sinking distance and overhanging surface roughness of square-shape and circle-shape inner structure can be found in Table 6, Table 7, Table 8, Table 9 and Table 10, respectively.
(3)$$D_{1s}=\left|\begin{array}{l} 1.90523+0.023433\mu -0.006618P-0.001047v-8.87778h-0.00032\mu P\\ +0.000055\mu v-0.125\mu h-0.0000013pv-0.010667Ph+0.00275vh\\ +0.00563333\mu^{2}+0.0000290133P^{2}+0.000000350833v^{2}+23.89815h^{2} \end{array}\right|$$
(4)$$\begin{array}{l} R_{1s}=127.38333+3.24250\mu -0.2015P-0.1053v-471.80556h-0.0014\mu P\\ +0.001225\mu v-8.16667\mu h-0.000013pv-0.98Ph+0.2025vh+0.32333\mu^{2}\\ +0.00110533P^{2}+0.0000308333v^{2}+1259.25926h^{2} \end{array}$$
(5)$$D_{2c}=\left|\begin{array}{l} 1.57439+0.069775\mu -0.00849867P-0.000365708v-7.475h-0.00035\mu P\\ +0.00002625\mu v-0.19167\mu h-0.0000000000000000000143982pv\\ -0.00566667Ph+0.00108333vh+0.005825\mu^{2}+0.00002752P^{2}+\\ 0.000000076875v^{2}+22.44444h^{2} \end{array}\right|$$
(6)$$\begin{array}{l} R_{2c}=77.7404+6.5315\mu -0.2004P-0.044195v-247.80556h-0.0006\mu P\\ -0.00175\mu v-9.58333\mu h-0.0000695pv-0.9Ph+0.051667vh+1.44575\mu^{2}\\ +0.0011812P^{2}+0.0000180812v^{2}+1139.72222h^{2} \end{array}$$

In Table 6, Table 7, Table 8 and Table 9, p represents the model confidence while *F* represents the result significance of the predicted model. The variance results showed that the *p* values were all lower than 0.0001, which indicted the reliability of the model. As for the *F* value, it can be seen that all of these data were higher than 0.05 which means the non-significant lack of fit of the prediction model. This conclusion further verified the reliability of the predicted model. The residual normal distribution and predicted-actual results of square-shape and circle-shape inner structure showed that most of the measured data were about the fitting line and presented a relative stable linear trend shown in Figure 5 and Figure 6. The predictive *R*-Squared of sinking distance for the circle-shape inner structure was 0.9367 which was in reasonable agreement with the adjusted *R*-Squared of 0.9751, while the predictive *R*-Squared of overhanging surface roughness for the circle-shape inner structure was 0.5219 which was not so close compared to the adjusted *R*-Squared of 0.8217. It may be caused by the random powder bonding on the overhanging surface which had a much more significant impact compared to the sinking distance of the circle-shape inner structure. The predictive *R*-Squared of sinking distance for the circle-shape inner structure was 0.8108 which was in reasonable agreement with the adjusted *R*-Squared of 0.9294, while the predictive *R*-Squared of overhanging surface roughness for the circle-shape inner structure was 0.9364 which was also in reasonable agreement with the adjusted *R*-Squared of 0.9767. The 3D surface plot further verified the results gained above shown in Figure 7, Figure 8, Figure 9 and Figure 10. To sum it up, the above results confirmed the reliability of the prediction model.

From Table 6, Table 7, Table 8 and Table 9, it can be found that laser power and layer thickness showed a quite significant impact on the sinking distance and overhanging surface roughness of square-shape inner structure while defocusing amount and scan speed showed less impact compared to the above-mentioned process parameters. As for the circle-shape inner structure, it can be found that all four parameters had a significant impact on sinking distance while overhanging surface roughness was affected dramatically by laser power and layer thickness. To have a better understanding on the phenomenon we have observed herein, the different formation mechanism of circle-shape and square-shape inner structure as discussed here is shown in Figure 11 and Figure 12.

It can be seen that the lack of support was the main reason resulting in the sinking of the overhanging surface on the square-shape inner structure while the powder bonding was the most obvious factor affecting the sinking distance of the circle-shape inner structure. As for the overhanging surface roughness, it can be found that the sinking of the molten pool and the bonding of the powder were two main factors which led to the increasing of square-shape overhanging surface roughness while it was more complicated on circle-shape overhanging surface roughness. Besides the factors we have mentioned above, the filling of the bonding powder on overhanging surface may even lead to lower surface roughness values, as shown in Figure 12.

Based on the results mentioned above, genetic algorithm was employed in this work to search for an optimal solution. The Gamultiobj function was one of the widely used algorithm in all these genetic algorithms. This function was improved using the NSGA-2 method with the help of Matlab. The relative optimize flow is shown in Figure 13.

Pareto optimal process parameter solution set was calculated accordingly. As so many parameter combinations were gained here, only 50 groups (25 groups for square-shape inner structure and 25 groups for circle-shape inner structure) are listed in this paper, as shown in Table 10.

The Pareto front optimization results revealed that the defocusing amount and scan speed was better fluctuating around −1.50 mm and 1196 mm/s, respectively, while the laser power and layer thickness ranged within 160 W to 165 W and 0.150 mm to 0.152 mm separately for square-shape overhanging surface quality. As for circle-shape overhanging surface quality, it can be seen that the sinking distance and overhanging surface roughness was hard to guarantee at the same time. When the defocusing amount fluctuated around −1.60 mm to −1.65 mm, the laser power went higher while the scan speed decreased at the same time, increasing the laser energy input. This resulted in the increasing of the sinking distance while overhanging surface roughness showed a significant decreasing trend. When the defocusing amount ranged around −1.45 mm to −1.50 mm, laser power and scan speed showed an opposite trend compared to higher defocusing amount and the sinking distance had a quite significant improvement while surface roughness value increased at the same time. The above results on circle-shape overhanging surface showed that two main directions for overhanging surface quality improvement can be employed: sinking distance optimization and surface roughness optimization. This result further proved the forming mechanism we have previously determined.

To further verify the accuracy of the results listed above, experiments were conducted according to the process combination, and the deviation rate was employed to study the accuracy of the prediction results as shown in Equation (7).
(7)$$\alpha =\frac{\left|\omega_{1}-\omega_{0}\right|}{\omega_{0}}$$

In this equation, α represents deviation rate, $\omega_{1}$ represents measured data and $\omega_{0}$ represents prediction data. The deviation rate of experiment and optimization results of square-shape and circle-shape overhanging surface quality can be seen in Table 11.

From the testing results, it can be found that the error ranged within 10%, which was mainly caused by random powder bonding and the process parameters simplification in printing process. Taking both of these two hard-to-control factors into account, 10% deviation rate in this work was quite reasonable and acceptable. To further verify the results, the morphology of the overhanging surface was observed using SEM, as shown in Figure 14.

Although the sinking of the overhanging surface was still obvious, it showed a quite obvious improvement compared to the overhanging surface quality printed before. This further verified the results we have obtained in this research.

## 4. Conclusions

In this work, process parameter combination, including defocusing amount, laser power, scan speed and layer thickness, was optimized to increase the overhanging surface quality of square-shape and circle-shape inner structure using genetic algorithm. The main findings are listed as follows.

The sinking distance and overhanging surface roughness of the square-shape inner structure showed a significant downward trend printed using optimized process parameter combination compared to the parameter combination used before.The circle-shape inner structure had two obvious optimization directions, sinking distance and overhanging surface roughness, which are mainly caused by the different forming mechanism in the printing process compared to the square-shape overhanging surface. This was the first study to compare the formation mechanism of the circle- and square-shape inner structures as far as we know.According to the optimization results, it can be seen that the defocusing amount and scan speed was better fluctuating around −1.50 mm and 1196 mm/s, respectively, while the laser power and layer thickness ranged within 160 W to 165 W and 0.150 mm to 0.152 mm, separately, for the square-shape overhanging surface quality.When the defocusing amount fluctuated around −1.60 mm to −1.65 mm, the laser power went higher while the scan speed decreased at the same time, increasing the laser energy input. This resulted in an increase in the sinking distance while overhanging surface roughness showed a significant decreasing trend for the circle-shape overhanging surface quality. When the defocusing amount ranged around −1.45 mm to −1.50 mm, laser power and scan speed showed an opposite trend compared to higher defocusing amount and the sinking distance had a quite significant improvement while surface roughness value increased at the same time.The difference of experiment data used for verification ranged within 10% compared to the computational results which further proved the reliability of the optimization gained in this work.

## Figures and Tables

**Figure 1 materials-15-04607-f001:**
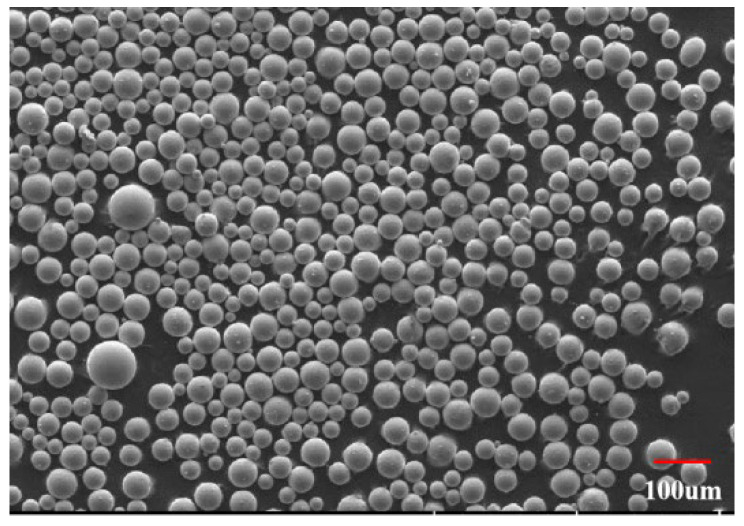
TC4 powder used in this work.

**Figure 2 materials-15-04607-f002:**
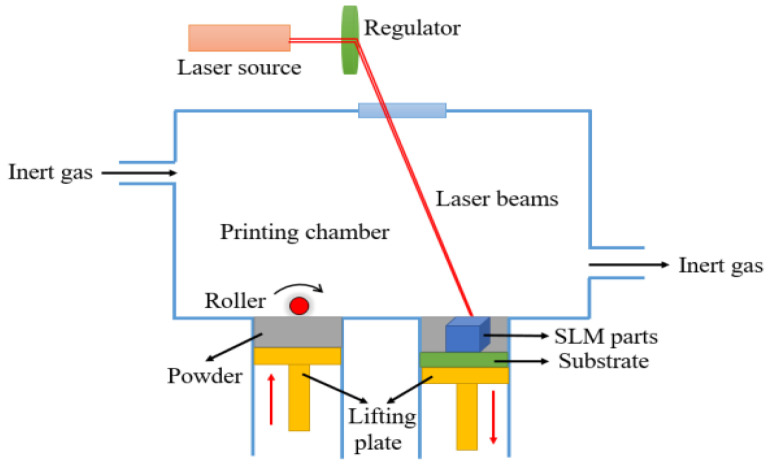
Schematic diagram of SLM.

**Figure 3 materials-15-04607-f003:**
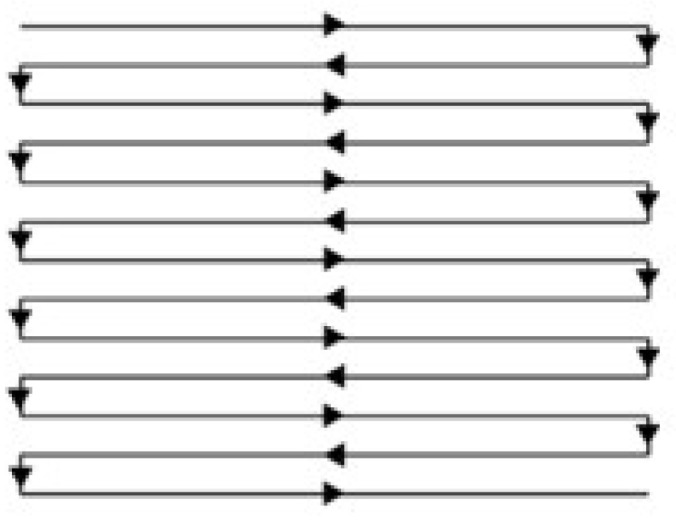
Z-shape scan strategy used in this work.

**Figure 4 materials-15-04607-f004:**
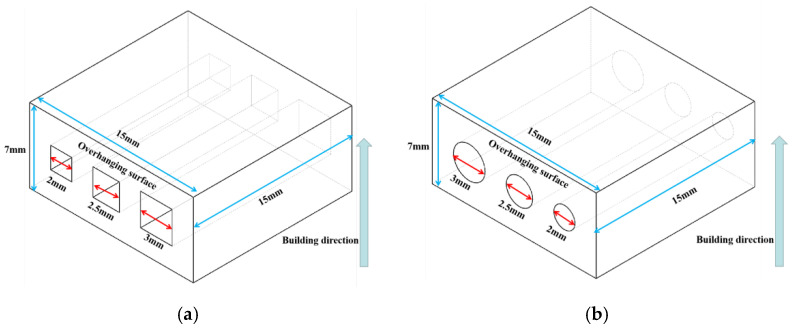
Designed model (**a**,**b**) and printed parts (**c**).

**Figure 5 materials-15-04607-f005:**
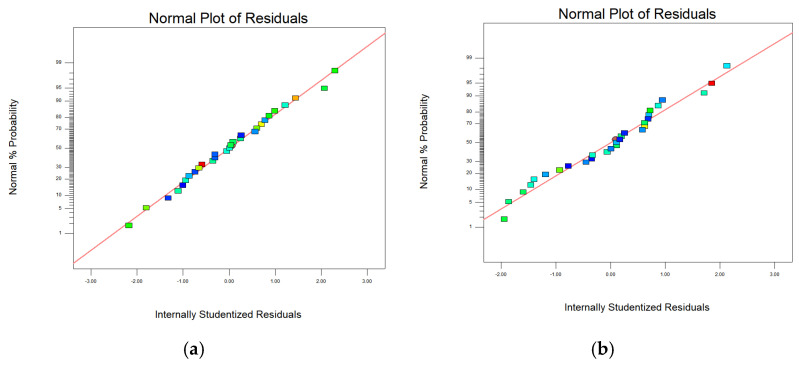
Residual normal distribution on sinking distance and overhanging surface roughness of square-shape (**a**,**b**) and circle-shape (**c**,**d**) inner structure.

**Figure 6 materials-15-04607-f006:**
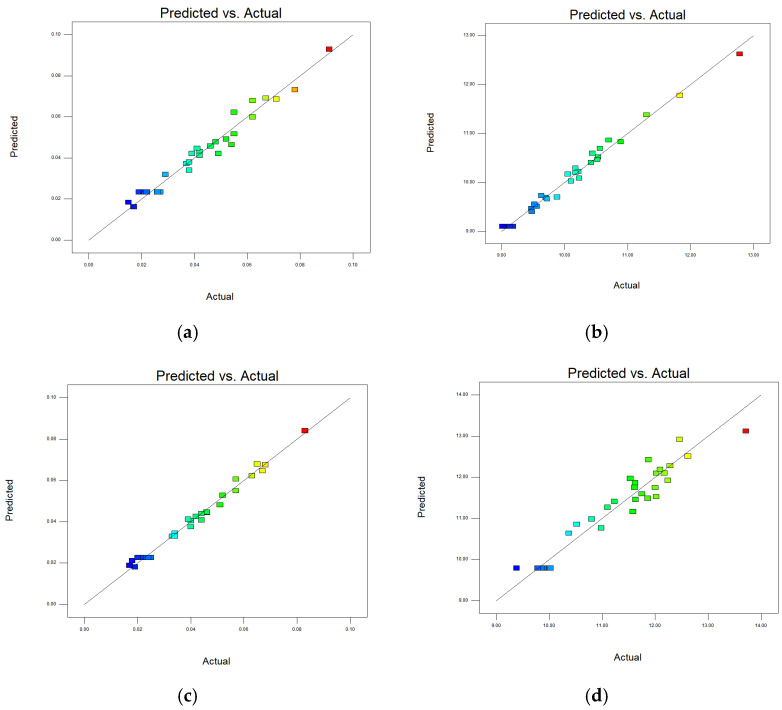
Comparison of predicted and actual results on sinking distance and overhanging surface roughness of square-shape (**a**,**b**) and circle-shape (**c**,**d**) inner structure.

**Figure 7 materials-15-04607-f007:**
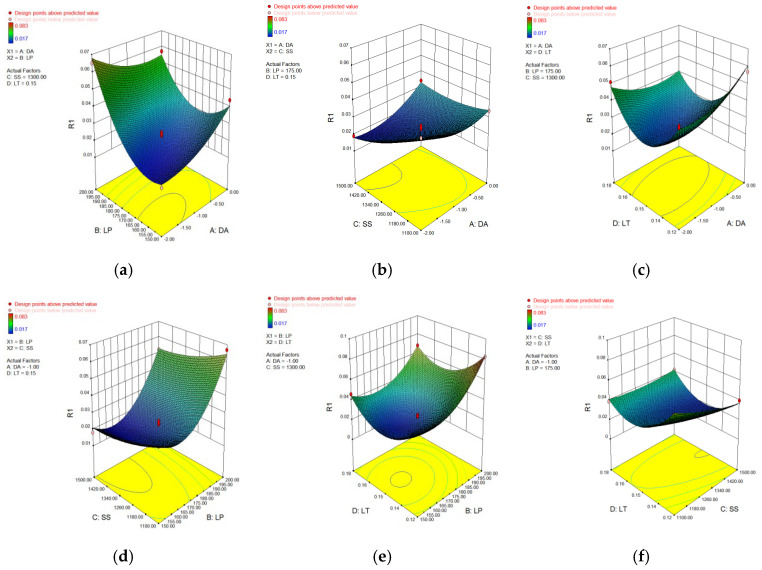
3D surface plot of defocusing amount/laser power (**a**), defocusing amount/scan speed (**b**), defocusing amount/layer thickness (**c**), laser power/scan speed (**d**), laser power/layer thickness (**e**) and scan speed/layer thickness (**f**) of sinking distance of circle-shape inner structure.

**Figure 8 materials-15-04607-f008:**
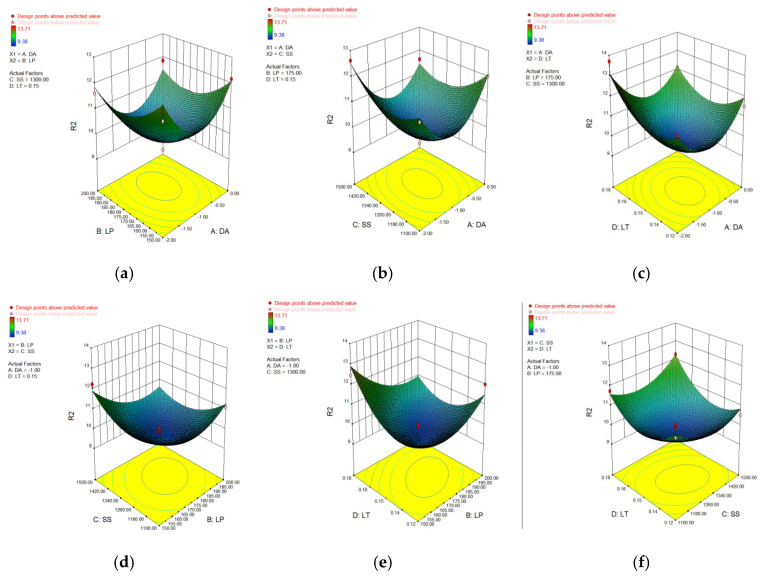
3D surface plot of defocusing amount/laser power (**a**), defocusing amount/scan speed (**b**), defocusing amount/layer thickness (**c**), laser power/scan speed (**d**), laser power/layer thickness (**e**) and scan speed/layer thickness (**f**) of overhanging surface roughness of circle-shape inner structure.

**Figure 9 materials-15-04607-f009:**
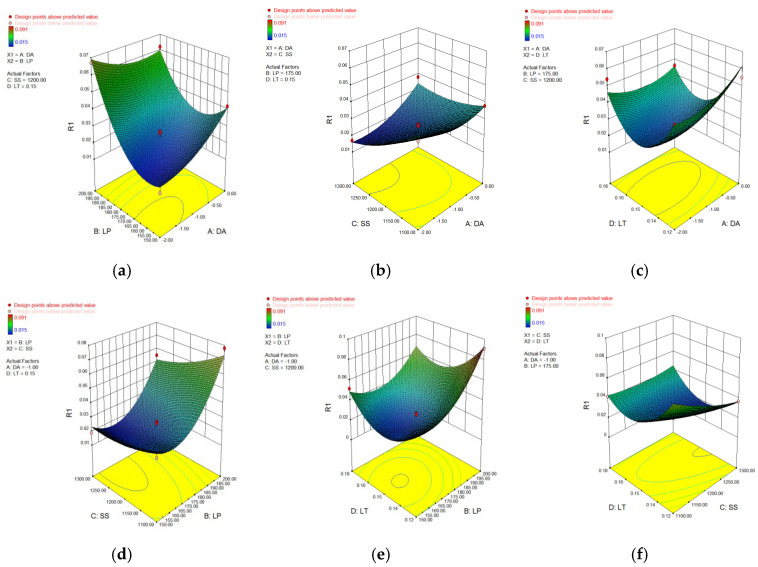
3D surface plot of defocusing amount/laser power (**a**), defocusing amount/scan speed (**b**), defocusing amount/layer thickness (**c**), laser power/scan speed (**d**), laser power/layer thickness (**e**) and scan speed/layer thickness (**f**) of sinking distance of square-shape inner structure.

**Figure 10 materials-15-04607-f010:**
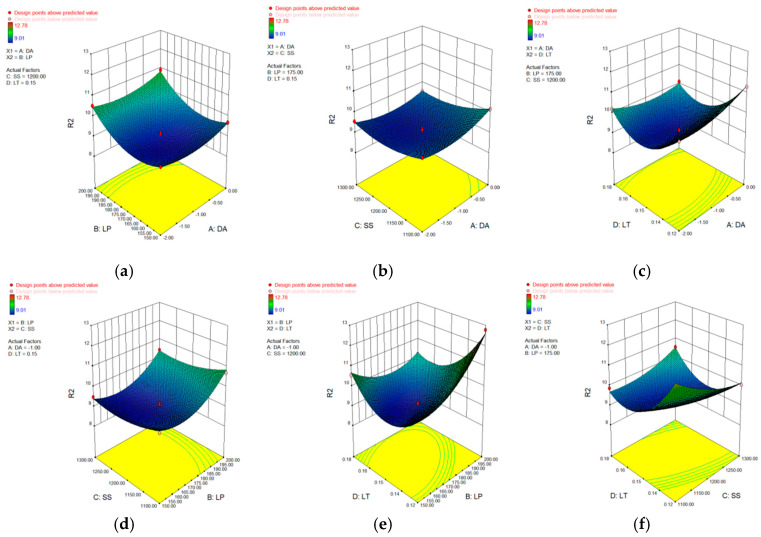
3D surface plot of defocusing amount/laser power (**a**), defocusing amount/scan speed (**b**), defocusing amount/layer thickness (**c**), laser power/scan speed (**d**), laser power/layer thickness (**e**) and scan speed/layer thickness (**f**) of overhanging surface roughness of square-shape inner structure.

**Figure 11 materials-15-04607-f011:**
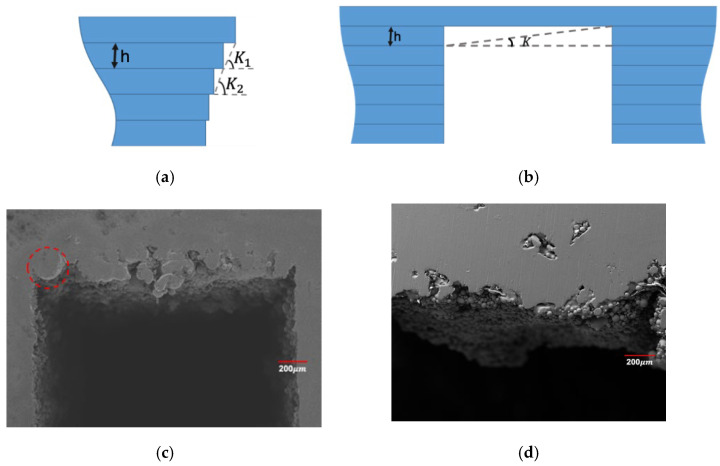
Amplified images on the edge of the circle- (**a**) and square-shape (**b**) inner structure and morphology of square-shape overhanging surface (**c**,**d**).

**Figure 12 materials-15-04607-f012:**
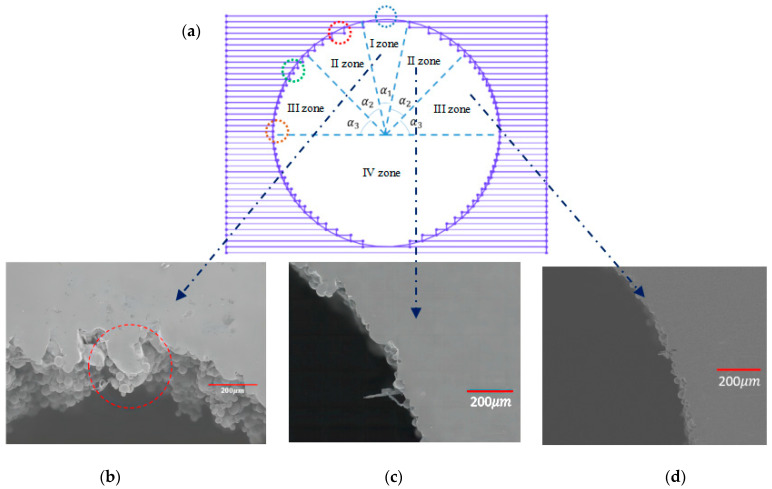
Schematic diagram on the printing of circle-shape inner structure part (**a**) and the morphology of zone I (**b**), zone II (**c**) and zone III (**d**).

**Figure 13 materials-15-04607-f013:**
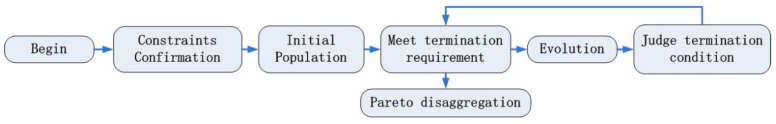
Flow chart of multi-objectives genetic algorithm.

**Figure 14 materials-15-04607-f014:**
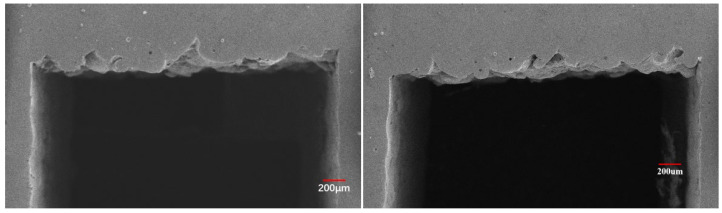
Morphology of the overhanging surface printed using optimized process parameter combination.

**Table 1 materials-15-04607-t001:** Relative information about the TC4 powder after processing.

Powder	Sphericity	Flowability/s	D_10_/μm	D_50_/μm	D_90_/μm
Ti6Al4V	0.982	12.3	16.3	18.5	21.2

**Table 2 materials-15-04607-t002:** Process parameters used in this work.

	Defocusing Amount	Laser Power	Scan Speed	Layer Thickness	Inert Gas	Hatch Spacing
Circle-shape	−2.0 mm ≤ *μ* ≤ 0.0 mm	150 W ≤ *P* ≤ 200 W	1100 mm/s ≤ *v* ≤ 1500 mm/s	0.12 mm ≤ *h* ≤ 0.18 mm	Argon	0.10 mm
Square-shape	−2.0 mm ≤ *μ* ≤ 0.0 mm	150 W ≤ *P* ≤ 200 W	1100 mm/s ≤ *v* ≤ 1300 mm/s	0.12 mm ≤ *h* ≤ 0.18 mm	Argon	0.10 mm

**Table 3 materials-15-04607-t003:** Factor level table of square-shape and circle-shape inner structure.

Factor/Level		−1	0	1
Square-shape inner structure	Defocusing amount *μ* (mm)	−2.0	−1.0	0.0
	Laser power *P* (W)	150	175	200
	Scan speed *v* (mm/s)	1100	1200	1300
	Layer thickness *h* (mm)	0.12	0.15	0.18
Circle-shape inner structure	Defocusing amount *μ* (mm)	−2.0	−1.0	0.0
	Laser power *P* (W)	150	175	200
	Scan speed *v* (mm/s)	1100	1300	1500
	Layer thickness *h* (mm)	0.12	0.15	0.18

**Table 4 materials-15-04607-t004:** Experimental data of the square-shape inner structure part.

	*μ* (mm)	*P* (W)	*v* (mm/s)	*h* (mm)	Sinking Distance *D*_1*s*_/mm	Surface Roughness *R*_1*s*_/μm
1	−1.0	175	1200	0.15	0.02	9.06
2	−1.0	150	1300	0.15	0.019	9.48
3	0.0	150	1200	0.15	0.042	9.7
4	−2.0	175	1300	0.15	0.017	9.56
5	0.0	175	1100	0.15	0.038	10.17
6	−1.0	175	1100	0.18	0.042	9.88
7	−1.0	175	1200	0.15	0.022	9.18
8	−2.0	150	1200	0.15	0.015	9.47
9	−2.0	200	1200	0.15	0.067	10.52
10	0.0	200	1200	0.15	0.062	10.89
11	−1.0	150	1100	0.15	0.029	9.63
12	−1.0	175	1300	0.12	0.037	10.05
13	−1.0	175	1100	0.12	0.071	11.83
14	0.0	175	1200	0.18	0.046	10.1
15	−2.0	175	1200	0.18	0.054	10.22
16	−1.0	175	1200	0.15	0.022	9.13
17	−1.0	175	1300	0.18	0.041	10.53
18	−1.0	200	1100	0.15	0.078	10.7
19	−2.0	175	1100	0.15	0.039	9.72
20	0.0	175	1200	0.12	0.055	11.3
21	0.0	175	1300	0.15	0.038	9.52
22	−1.0	200	1300	0.15	0.055	10.42
23	−1.0	150	1200	0.18	0.052	10.56
24	−1.0	200	1200	0.18	0.062	10.17
25	−1.0	150	1200	0.12	0.049	10.23
26	−1.0	200	1200	0.12	0.091	12.78
27	−2.0	175	1200	0.12	0.048	10.44
28	−1.0	175	1200	0.15	0.027	9.01
29	−1.0	175	1200	0.15	0.026	9.12

**Table 5 materials-15-04607-t005:** Experimental data of the circle-shape inner structure part.

	*μ* (mm)	*P* (W)	*v* (mm/s)	*h* (mm)	Sinking Distance *D*_2*c*_/mm	Surface Roughness *R*_2*c*_/μm
1	−1.0	175	1100	0.12	0.063	11.23
2	−1.0	150	1300	0.18	0.046	12.46
3	−1.0	175	1100	0.18	0.039	11.75
4	−1.0	150	1300	0.12	0.044	10.98
5	−1.0	175	1300	0.15	0.021	9.78
6	−1.0	150	1500	0.15	0.018	12.24
7	−1.0	200	1100	0.15	0.067	11.1
8	−1.0	175	1300	0.15	0.023	9.9
9	0.0	175	1500	0.15	0.034	11.63
10	0.0	200	1300	0.15	0.057	11.86
11	−1.0	175	1500	0.12	0.04	10.52
12	−2.0	175	1300	0.18	0.051	13.71
13	−2.0	200	1300	0.15	0.065	11.62
14	0.0	175	1300	0.18	0.039	12.09
15	−1.0	175	1500	0.18	0.042	12.28
16	0.0	150	1300	0.15	0.044	12.17
17	−1.0	200	1300	0.12	0.083	12.02
18	−2.0	175	1500	0.15	0.019	12.62
19	−1.0	200	1300	0.18	0.068	10.8
20	−1.0	175	1300	0.15	0.024	9.38
21	−2.0	175	1100	0.15	0.04	11.61
22	−1.0	150	1100	0.15	0.033	11.58
23	−2.0	175	1300	0.12	0.046	12
24	−1.0	200	1500	0.15	0.052	10.37
25	−1.0	175	1300	0.15	0.02	10.02
26	−1.0	175	1300	0.15	0.025	9.85
27	0.0	175	1100	0.15	0.034	12.02
28	−2.0	150	1300	0.15	0.017	11.87
29	0.0	175	1300	0.12	0.057	11.53

**Table 6 materials-15-04607-t006:** Analysis of variance in regression results of sinking distance for square-shape overhanging surface.

Variation Source	Quadratic Sum	DOF	Mean Square	*F* Value	*p* Value
Model	0.010	14	7.206 × 10^−4^	27.33	<0.0001 (significant)
μ	1.401 × 10^−4^	1	1.401 × 10^−4^	5.31	0.0370
P	3.640 × 10^−3^	1	3.640 × 10^−3^	138.06	<0.0001
v	6.750 × 10^−4^	1	6.750 × 10^−4^	25.60	0.0002
h	2.430 × 10^−4^	1	2.430 × 10^−4^	9.22	0.0089
μP	2.560 × 10^−4^	1	2.560 × 10^−4^	9.71	0.0076
μv	1.210 × 10^−4^	1	1.210 × 10^−4^	4.59	0.0502
μh	5.625 × 10^−5^	1	5.625 × 10^−5^	2.13	0.1662
Pv	4.225 × 10^−5^	1	4.225 × 10^−5^	1.60	0.2262
Ph	2.560 × 10^−4^	1	2.560 × 10^−4^	9.71	0.0076
vh	2.722 × 10^−4^	1	2.722 × 10^−4^	10.33	0.0063
$\mu^{2}$	2.058 × 10^−4^	1	2.058 × 10^−4^	7.81	0.0143
$P^{2}$	2.133 × 10^−3^	1	2.133 × 10^−3^	80.90	<0.0001
$v^{2}$	7.984 × 10^−5^	1	7.984 × 10^−5^	3.03	0.1038
$h^{2}$	3.001 × 10^−3^	1	3.001 × 10^−3^	113.81	<0.0001
Residual	3.691 × 10^−4^	14	2.637 × 10^−5^		
Lack of fit	3.339 × 10^−4^	10	3.339 × 10^−5^	3.79	0.1053 (non−significant)
Pure error	3.520 × 10^−5^	4	8.800 × 10^−6^		
Sum of square	0.010	28			

**Table 7 materials-15-04607-t007:** Analysis of variance in regression results of roughness for square-shape overhanging structure.

Variation Source	Quadratic Sum	DOF	Mean Square	*F* Value	*p* Value
Model	20.19	14	1.44	84.72	<0.0001 (significant)
μ	0.26	1	0.26	14.99	0.0017
P	3.42	1	3.42	201.17	<0.0001
v	0.47	1	0.47	27.50	0.0001
h	2.23	1	2.23	130.87	<0.0001
μP	4.900 × 10^−3^	1	4.900 × 10^−3^	0.29	0.6000
μv	0.060	1	0.060	3.53	0.0814
μh	0.24	1	0.24	14.11	0.0021
Pv	4.225 × 10^−3^	1	4.225 × 10^−3^	0.25	0.6261
Ph	2.16	1	2.16	126.96	<0.0001
vh	1.48	1	1.48	86.73	<0.0001
$\mu^{2}$	0.68	1	0.68	39.84	<0.0001
$P^{2}$	3.10	1	3.10	181.88	<0.0001
$v^{2}$	0.62	1	0.62	36.23	<0.0001
$h^{2}$	8.33	1	8.33	489.51	<0.0001
Residual	0.24	14	0.017		
Lack of fit	0.22	10	0.022	5.08	0.0656 (non-significant)
Pure error	0.017	4	4.350 × 10^−3^		
Sum of square	20.43	28			

**Table 8 materials-15-04607-t008:** Analysis of variance in regression results of the dimension error for circle-shape overhanging surface.

Variation Source	Quadratic Sum	DOF	Mean Square	*F* Value	*p* Value
Model	8.380 × 10^−3^	14	5.985 × 10^−4^	79.21	<0.0001 (significant)
μ	6.075 × 10^−5^	1	6.075 × 10^−5^	8.04	0.0132
P	3.008 × 10^−3^	1	3.008 × 10^−3^	398.14	<0.0001
v	4.201 × 10^−4^	1	4.201 × 10^−4^	55.60	<0.0001
h	1.920 × 10^−4^	1	1.920 × 10^−4^	25.41	0.0002
μP	3.063 × 10^−4^	1	3.063 × 10^−4^	40.53	<0.0001
μv	1.103 × 10^−4^	1	1.103 × 10^−4^	14.59	0.0019
μh	1.323 × 10^−4^	1	1.323 × 10^−4^	17.50	0.0009
Pv	0.000	1	0.000	0.000	1.0000
Ph	7.225 × 10^−5^	1	7.225 × 10^−5^	9.56	0.0080
vh	1.690 × 10^−4^	1	1.690 × 10^−4^	22.37	0.0003
$\mu^{2}$	2.201 × 10^−4^	1	2.201 × 10^−4^	29.13	<0.0001
$P^{2}$	1.919 × 10^−3^	1	1.919 × 10^−3^	253.97	<0.0001
$v^{2}$	6.133 × 10^−5^	1	6.133 × 10^−5^	8.12	0.0129
$h^{2}$	2.647 × 10^−3^	1	2.647 × 10^−3^	350.29	<0.0001
Residual	1.058 × 10^−4^	14	7.556 × 10^−6^		
Lack of fit	8.858 × 10^−5^	10	8.858 × 10^−6^	2.06	0.2534 (non-significant)
Pure error	1.720 × 10^−5^	4	4.300 × 10^−6^		
Sum of square	8.485 × 10^−3^	28			

**Table 9 materials-15-04607-t009:** Analysis of variance in regression results of the surface roughness for circle-shape inner structure.

Variation Source	Quadratic Sum	DOF	Mean Square	*F* Value	*p* Value
Model	25.62	14	1.83	10.22	<0.0001 (significant)
μ	0.38	1	0.38	2.11	0.1682
P	1.04	1	1.04	5.80	0.0304
v	0.011	1	0.011	0.064	0.8044
h	1.93	1	1.93	10.77	0.0055
μP	9.000 × 10^−4^	1	9.000 × 10^−4^	5.026 × 10^−3^	0.9445
μv	0.49	1	0.49	2.74	0.1203
μh	0.33	1	0.33	1.85	0.1957
Pv	0.48	1	0.48	2.70	0.1228
Ph	1.82	1	1.82	10.18	0.0065
vh	0.38	1	0.38	2.15	0.1650
$\mu^{2}$	13.56	1	13.56	75.72	<0.0001
$P^{2}$	3.54	1	3.54	19.74	0.0006
$v^{2}$	3.39	1	3.39	18.95	0.0007
$h^{2}$	6.82	1	6.82	38.12	<0.0001
Residual	2.51	14	0.18		
Lack of fit	2.27	10	0.23	3.84	0.1035 (non-significant)
Pure error	0.24	4	0.059		
Sum of square	28.12	28			

**Table 10 materials-15-04607-t010:** Pareto optimal front of multi-objective optimization for the process parameter of square-shape inner structure and circle-shape inner structure.

	Number	μ/mm	P/W	$v/\mathrm{mm}\cdot s^{-1}$	h/mm	$D/{mm}_{\;}$	R/μm
Square-shape inner structure	1	−1.677	160.481	1195.741	0.148	0.017	9.086
	2	−1.497	166.279	1195.907	0.154	0.018	9.023
	3	−1.675	160.504	1195.746	0.148	0.017	9.086
	4	−1.674	161.389	1195.867	0.150	0.017	9.080
	5	−1.642	167.085	1195.869	0.155	0.019	9.069
	6	−1.188	166.122	1196.201	0.152	0.019	8.986
	7	−1.643	163.485	1195.985	0.151	0.017	9.060
	8	−1.445	165.546	1196.050	0.152	0.018	9.008
	9	−1.521	160.984	1195.774	0.149	0.017	9.044
	10	−1.565	163.057	1195.823	0.150	0.017	9.038
	11	−1.294	165.937	1196.183	0.152	0.018	8.989
	12	−1.416	166.230	1196.074	0.153	0.018	9.004
	13	−1.329	165.391	1196.197	0.151	0.018	8.992
	14	−1.638	166.817	1195.932	0.155	0.019	9.068
	15	−1.646	167.187	1195.845	0.155	0.019	9.078
	16	−1.585	166.257	1195.805	0.154	0.019	9.045
	17	−1.650	161.215	1195.803	0.150	0.017	9.074
	18	−1.519	161.015	1195.855	0.150	0.017	9.044
	19	−1.621	163.309	1195.840	0.152	0.018	9.059
	20	−1.471	161.159	1195.906	0.149	0.017	9.033
	21	−1.646	167.187	1195.845	0.155	0.019	9.078
	22	−1.316	164.598	1195.955	0.150	0.018	8.995
	23	−1.639	160.602	1195.788	0.150	0.017	9.077
	24	−1.569	166.297	1195.915	0.155	0.019	9.049
	25	−1.467	165.966	1195.875	0.152	0.018	9.012
Circle-shape inner structure	1	−1.621	161.007	1373.980	0.175	0.026	9.731
	2	−1.484	150.168	1425.044	0.169	0.029	11.513
	3	−1.632	167.124	1371.377	0.177	0.012	11.469
	4	−1.487	150.541	1414.627	0.166	0.011	11.415
	5	−1.552	153.802	1389.198	0.168	0.009	12.600
	6	−1.621	165.402	1373.357	0.176	0.010	12.190
	7	−1.484	150.185	1423.787	0.164	0.010	12.292
	8	−1.636	167.580	1371.051	0.177	0.009	12.514
	9	−1.628	163.245	1371.910	0.176	0.009	12.693
	10	−1.492	150.439	1423.250	0.166	0.009	12.638
	11	−1.557	161.059	1388.061	0.175	0.009	12.462
	12	−1.555	155.022	1389.417	0.168	0.011	11.702
	13	−1.540	162.648	1380.916	0.174	0.027	10.958
	14	−1.633	167.446	1371.084	0.177	0.009	12.693
	15	−1.545	152.729	1398.116	0.165	0.010	11.879
	16	−1.635	167.580	1371.052	0.177	0.022	9.792
	17	−1.496	150.500	1417.266	0.167	0.024	10.402
	18	−1.635	166.765	1371.397	0.176	0.009	12.650
	19	−1.582	163.643	1381.478	0.174	0.020	9.844
	20	−1.632	166.067	1371.503	0.176	0.015	10.650
	21	−1.625	166.003	1374.742	0.175	0.019	9.974
	22	−1.491	150.945	1408.666	0.168	0.010	12.154
	23	−1.530	150.508	1409.174	0.174	0.020	9.933
	24	−1.498	153.907	1414.584	0.167	0.014	10.704
	25	−1.630	165.668	1371.567	0.176	0.027	11.015

**Table 11 materials-15-04607-t011:** Deviation rate of experiment and optimization results of square-shape and circle-shape overhanging surface quality.

	μ/mm	P/W	$v/\mathrm{mm}\cdot s^{-1}$	h/mm	$Deviation_{D}/{\%}_{\;}$	$Deviation_{R}/\mu m$	Prediction
Square-shape inner structure	−1.40	165	1200	0.15	8.75	6.77	12
	−1.50	160	1200	0.15	9.45	8.54	18
Circle-shape inner structure	−1.55	150	1400	0.17	10.07	8.62	2
	−1.60	165	1350	0.18	9.65	7.87	6

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
