# Peer review of "A Multi-Objectives Genetic Algorithm Based Predictive Model and Strategy Optimization during SLM Process"

_materials, 2022, doi:10.3390/ma15134607_

Round 1

Reviewer 1 Report

The work investigated on the influence of building strategies on the surface quality of SLM processed parts. However, following shortcomings are presented in the manuscript and which has to be rectified before further processing. I request mandatory revision, as listed below, please do not simply respond but revise manuscript.

·         An extensive language editing is required to improve the quality of the manuscript. Since the article has several language errors, therefore, the authors are urged to rewrite the manuscript with a native English speaker.

·         Numerous research articles were already published as such specific investigations on SLM process. Authors should mention how this work is differentiated from the existing studies.

·         Introduction, especially literature survey is not sufficient for problem definition. Include similar recent research works in this section.

·         For readers to quickly catch the contribution in this work, it would be better to highlight major difficulties and challenges, and authors' original achievements to overcome them, in a clearer way in Introduction section.

·         A detailed Experimental procedure should be incorporated including the SEM microstructure of the powder particles, powder particle size distribution, etc.

·         It is suggested to highlight the limitations of this study, suggested improvements of this work and future directions in the conclusion section. Also, the conclusion can be presented better than the present form with more findings.

·         On what basis the processing parameters and their levels has been selected for this investigation. Unfortunately, there are no obvious justification was found in the article.

·         The equations should be written in Mathtype® software instead of MS Equation.

·         Authors have mentioned that they have performed 29 experimental runs, however, the Table 4&5 shows only 10 experiments. It should be explained detailly.

·         Why the experiments were performed by random sets? There are numerous DOE strategies such as RSM, Factorial and OA are available in Design expert software. A specific justification should be incorporated in the manuscript for the selected DOE strategy.

·         The statistical analysis of performed experiments were justified with Co-efficient of Determination (R2) along with ANOVA. But, the authors have not mentioned any R2 value in order to justify their experiment’s efficiency.

·         Since, the investigation involves several processing parameters, but the influence of these parameters on the selected response features were not discussed with the aid of 3D interaction plots. It should be included in order to scientifically investigate the impact of selected parameters.

·         The work utilized NSGA for multi-response optimization, however, the detailed explanation of how the optimization parameters were selected, how the objective functions were incorporated for optimization and pareto optimal front were missing. Which creates more concerns about the investigation and optimization?

·         More SEM images should be incorporated for different fabrication conditions in order to investigate the hanging nature of parts for different conditions.  

·         Moreover, the results and discussion are not clearly dealt the outcomes of the proposed work. The authors should explicitly state the novel contribution of this work, the similarities and the differences of this work with the previous publications in this section.

Please note that the comments are intended merely to assist the authors in improving the manuscript and ensuring that published papers are of the highest quality. They are in NO WAY intended to discourage or demean the authors personally.

Author Response

Dear Reviewer,

Thanks a lot for your help and consideration. The detail reply are listed in the attached file.

Reviewer 2 Report

Selective Laser Melting (SLM) is an emerging Laser Powder Bed Fusion (L-PBF) additive manufacturing (AM) process for producing prototypes and fully functional metallic products in a short period of time. Similar to other AM techniques, the SLM technology enables the production of parts that cannot be manufactured by conventional technologies such as machining and pressure casting. Specifically, it is possible to manufacture parts with curved internal cooling channels, lattice structures, and complex geometry shapes. The SLM technology also reduces the number of parts in assemblies to one complex design with all the necessary features and proper functions. Four main process parameters, including defocusing amount, laser power, scan speed, and layer thickness were studied in this work and an optimal process parameter combination was obtained wo improve circle-shape and square-shape overhanging surface quality respectively using multi-objective genetic algorithm. Experiments were also conducted at last to verify the prediction model and results gained before. The study seems to be carefully done and has been well reported with the exception of some issues detailed below.

- results and discussion and conclusion parts are inadequate according to citation and analyze in detail. There should be the importance of the study in detail, comparison results with other approaches in literature, the success of the prediction and computational results. 

- the authors however paid little attention to the review of the literature (few sources are given) and substantiation of the relevance of the presented study. It is not written in scientific perspective. This section should be widened, its present form too short. In this section, more literature papers have to be included to explain the subject better. Now the list of references needs to be supplemented with at least several more references published over the past 5 years. 

- the conclusions are too general, the main findings and highlights of the study may be summarised as follows,

- figure 1 (schematic diagram of SLM) needs to be moved to the Materials and Methods section,

- the technical features, measuring ranges and accuracy of the measuring devices are not stated (Low-speed Wire Cutting machine , Trilinear Coordinates Measuring instrument,  Roughometer,  Scanning Electron Microscope (SEM)

Author Response

Dear Reviewer, 

 Thank you for your time and consideration. The detail reply are listed in the attached file.

Reviewer 3 Report

The manuscript "A Multi-Objectives Genetic Algorithm Based Predictive Model and Strategy Optimization during SLM Process" reports on the relationship between process parameters and quality if different parts, the process was optimized using multi-objectives genetic algorithm.

From a general point of view, the topic of the manuscript is interesting and worth investigating. The manuscript is, generally, clear, well-written, well-organized, and timely. The figures are clear and appealing.

The introduction clearly states the aim of the work and sharply inserts the work within a well-focused scientific and technological framework of general interest.

The experimental approaches are clear, reliable, and strongly founded, even if the experimental section needs to be completed with some missing details. The worth of mention is the effort by the authors in the quantitative discussions of the results concerning the analysis. On the contrary, the discussion of the results concerning the detailed analysis can be improved.

Overall, I find an interesting and valuable manuscript that requires some minor improvements before publication:

1.     The entire article demands significant minor grammatical improvement.

2.     Please expand and improve the introduction section a little more.

3.    The authors should demonstrate or at least comment about scaling up difficulties or how to ease it is to scale up their technique.

4.  As the authors mentioned quality of different parts, please provide enough evidence by doing mechanical testing.

5.  Please provide details regarding how you authors calculated the surface roughness.

Author Response

(The authors gave the same response as above.)

Round 2

Reviewer 1 Report

Authors have not responded properly for the queries raised. Especially the following comments were not addressed properly. The following comments should be recitify for proceeding further.

1. Authors have mentioned that they have performed 29 experimental runs, however, the Table 4&5 shows only 10 experiments. It should be explained detailly.

2. Why the experiments were performed by random sets? There are numerous DOE strategies such as RSM, Factorial and OA are available in Design expert software. A specific justification should be incorporated in the manuscript for the selected DOE strategy.

3. The statistical analysis of performed experiments were justified with Co-efficient of Determination (R2) along with ANOVA. But, the authors have not mentioned any R2 value in order to justify their experiment’s efficiency.

4. Since, the investigation involves several processing parameters, but the influence of these parameters on the selected response features were not discussed with the aid of 3D interaction plots. It should be included in order to scientifically investigate the impact of selected parameters.

5. The work utilized NSGA for multi-response optimization, however, the detailed explanation of how the optimization parameters were selected, how the objective functions were incorporated for optimization and pareto optimal front were missing. Which creates more concerns about the investigation and optimization?

6. Moreover, the results and discussion are not clearly dealt the outcomes of the proposed work. The authors should explicitly state the novel contribution of this work, the similarities and the differences of this work with the previous publications in this section.

Authors are instructed for a mandatory revision, as listed above, please do not simply respond but revise manuscript for further proceedings.

Author Response

Dear reviewer, 

Thank you for your time and consideration. The detail reply had already attached below.

Reviewer 2 Report

Thank you for accepting my comments, the article has been improved and I agree with its publication in the submitted form.

Author Response

Thanks again for your help and consideration.

Jitai